# Motherhood and Treatment Outcome in Female Patients with Compulsive Buying–Shopping Disorder

**DOI:** 10.3390/ijerph19127075

**Published:** 2022-06-09

**Authors:** Gemma Mestre-Bach, Roser Granero, Gemma Casalé-Salayet, Fernando Fernández-Aranda, Astrid Müller, Matthias Brand, Mónica Gómez-Peña, Laura Moragas, Isabel Sánchez, Lucía Camacho-Barcia, Alejandro Villena, Milagros L. Lara-Huallipe, Susana Jiménez-Murcia

**Affiliations:** 1Facultad de Ciencias de la Salud, Universidad Internacional de La Rioja, 26006 Logroño, Spain; gemma.mestre@unir.net; 2Ciber Fisiopatología Obesidad y Nutrición (CIBERObn), Instituto de Salud Carlos III, 28015 Madrid, Spain; roser.granero@uab.cat (R.G.); ffernandez@bellvitgehospital.cat (F.F.-A.); isasanchez@bellvitgehospital.cat (I.S.); lcamacho@idibell.cat (L.C.-B.); 3Departament de Psicobiologia i Metodologia de les Ciències de la Salut, Universitat Autònoma de Barcelona, 08193 Barcelona, Spain; 4Psychoneurobiology of Eating and Addictive Behaviors Group, Neurosciences Programme, Bellvitge Biomedical Research Institute (IDIBELL), 08908 Barcelona, Spain; monicagomez@bellvitgehospital.cat (M.G.-P.); lmoragas@bellvitgehospital.cat (L.M.); milagroslizbeth.lh@gmail.com (M.L.L.-H.); 5CPB-Serveis de Salut Mental, 08013 Barcelona, Spain; gcasale@cpbssm.org; 6Department of Psychiatry, Bellvitge University Hospital, 08907 Barcelona, Spain; 7Department of Clinical Sciences, School of Medicine and Health Sciences, University of Barcelona, 08007 Barcelona, Spain; 8Department of Psychosomatic Medicine and Psychotherapy, Hannover Medical School, 30625 Hannover, Germany; mueller.astrid@mh-hannover.de; 9General Psychology: Cognition and Center for Behavioral Addiction Research (CeBAR), University of Duisburg-Essen, 47057 Duisburg, Germany; matthias.brand@uni-due.de; 10Erwin L. Hahn Institute for Magnetic Resonance Imaging, 45141 Essen, Germany; 11Unidad Sexología Clínica y Salud Sexual, Consulta Dr. Carlos Chiclana, 28003 Madrid, Spain; alejandrovillena@doctorcarloschiclana.com

**Keywords:** compulsive buying disorder, motherhood, women, cognitive behavioral therapy, compliance, relapse, dropout

## Abstract

Motherhood has been proposed as an internal facilitating factor for the recovery of women with mental disorders. However, at the same time, there are significant barriers that may be interfering with the access and adherence to treatment for these women. The present longitudinal study aimed to deepen the sociodemographic and clinical profile of women with children and compulsive buying–shopping disorder (CBSD), and to explore the association between motherhood and response to treatment. The total sample included 77 women with a diagnosis of CBSD (*n* = 49 mothers) who received cognitive behavioral therapy (CBT) for 12 weeks. No association between psychopathology and motherhood was observed. The group of mothers reported an older age of onset of the CBSD, a lower amount of money spent per compulsive-buying episode, and a higher likelihood of family support for the CBSD. Moreover, this group showed lower risk of relapse. The findings support the theoretical proposal that considers motherhood as an internal facilitating factor for recovery and treatment adherence of mothers with addictions.

## 1. Introduction

Motherhood is a complex state that has been underexplored in mental health. Initial studies show, on the one hand, that parenting may be considered as a very relevant motivation for seeking treatment and improving health behaviors in women [1,2]. Therefore, motherhood has been considered as an internal facilitating factor, given that most women with mental illnesses report the desire to be good mothers and to take care of their children in the best possible way [1]. Consequently, those mothers who are able to fully identify with the parenting role may be more likely to experience an effective recovery [3]. 

On the other hand, mothers with mental illness’s successful recovery usually includes multiple aspects, such as improved self-care, involvement in treatment programs, and reconstruction of one’s identity [4,5,6]. However, there are several specific barriers that mothers may face when striving for recovery. Mothers with mental disorders may have difficulties in accessing essential resources, such as maintaining housing, obtaining childcare, and finding and retaining employment [7]. In addition, many mothers avoid making their pathology public and, consequently, do not seek treatment due to the associated stigma, or even for fear of losing custody of their child [8]. In addition, the lack of gender-sensitive treatment and other facilitating resources, such as childcare, prevent some women from feeling supported and engaging in existing treatment programs [9,10]. Finally, it is necessary to consider various structural problems, such as the time needed for clinical appointments and the treatment costs [1].

Regarding addictions, few previous studies have evaluated the specific role of motherhood. They have suggested that motherhood may be a motivating factor in recovering from substance use disorder, although it may also be a barrier due to fear of possible negative consequences, such as the loss of custody of their child [11,12]. However, this aspect has not been extensively studied in behavioral addictions. Gavriel-Fried and Ajzenstadt [13] reported that, in the case of mothers with gambling disorder, they retain control of their maternal duties while, at the same time, engaging in gambling behavior. In fact, they tended to maintain gambling behaviors since, among other aspects, their maternal role was well covered, and their responsibilities were not neglected. Suchman, DeCoste, McMahon, and Dalton [14] observed that 71–75% of mothers attended their clinical appointments, and that the most common reasons for dropout were relapse, transportation problems, family issues, and relocation. In the case of compulsive buying–shopping disorder (CBSD), understood as a behavioral addiction characterized by repetitive, irresistible, and overpowering urges to purchase goods which consequently generate severe distress or interference, most individuals who have this clinical condition are women [15,16,17]. Therefore, motherhood could have a specific impact on the psychotherapeutic treatment response of women with CBSD. 

Multiple approaches have been proposed for CBSD, both via pharmacological (i.e., citalopram, escitalopram, and fluvoxamine) and psychological (i.e., psychodynamic psychotherapy, behavioral approaches, and cognitive-behavioral interventions) approaches. Of all of these interventions, cognitive-behavioral therapy (CBT) appears to be particularly effective in reducing the associated distress of CBSD, with long-lasting effects [18].

In order to explore the role of motherhood in CBSD treatment outcome, the present study had the following aims: (1) to compare the sociodemographic and clinical profile of mothers and childless women with CBSD; and (2) to determine whether motherhood has a positive or negative impact on the treatment response by assessing dropout, relapse, and therapy compliance. It was hypothesized that, as proposed by other authors [1], motherhood will be a facilitating factor for treatment response that will outweigh the barriers faced by women with children and CBSD. Therefore, compared to those who do not have any children, we hypothesize that mothers will obtain lower rates of dropout, relapse, as well as higher compliance.

## 2. Materials and Methods

### 2.1. Participants 

A sample of *n* = 77 treatment-seeking patients with CBSD from the Department of Psychiatry at a University Hospital, recruited between March 2005 and June 2015, was considered. Participants were voluntarily referred to the Gambling Disorder and Other Behavioral Addictions Unit through general practitioners or other healthcare professionals, such as psychiatrists, clinical psychologists, and social workers.

Participants with additional mental disorders (i.e., schizophrenia, bipolar disorder, or other psychotic disorders) or intellectual disability were excluded from the study. The screening was performed through a structured interview by experienced clinicians before the treatment. These same therapists carried out the CBT program. 

### 2.2. Treatment

#### 2.2.1. Main Elements of the Treatment Program

The CBT program protocol for pathological gambling and other behavioral addictions has been previously described in detail [19], and its short and medium-term effectiveness has been reported in gambling disorder [20], as well as in CBSD [21,22]. The individual intervention for CBSD consisted of 12 weekly 45-min CBT sessions for outpatients. The program covered different general topics, including psychoeducation about the disorder (diagnostic definition, vulnerability factors, biopsychosocial models, disorder development and stages, etc.), together with stimulus control, such as money management and other potential triggers of relapse and alternative and compensatory behaviors. CBT also included the replacement of CBSD-associated behaviors for new healthy ones, and cognitive restructuring focused on illustrating and rectifying false illusions of control over compulsive buying. During the intervention, clinicians also worked on self-reinforcement, skills training, and relapse prevention techniques. Moreover, the program included exposure with response prevention (ERP) sessions [23], with the objective of reducing the patients’ arousal levels when exposed to certain stimuli which trigger the urge to enact buying behavior, and the intention of improving self-efficacy expectations for their CBSD recovery. The procedure consisted in confronting the patients in vivo with stimuli or situations that trigger the urge to buy (exposure) and prevent them from carrying out the behavior (response prevention). The ERP sessions were carried out in familiar shopping places, such as malls and commercial areas with a high density of preferred boutiques, as well as their favored shopping websites, where the subjects had to spend the mandated time in the presence of the activating stimulus. If the urge to buy did not diminish in this period of time, patients were advised to wait. For the first session of EPR, patients were advised to be accompanied in order to monitor activation levels, and, once these diminished, they continued with the consecutive sessions on their own. 

#### 2.2.2. Main Outcome Variables

The full recovery outcome, as the main objective of the treatment, was defined as the absence of compulsive buying episodes and the stabilization of other areas of the patients’ life, including emotional, family, professional, and social areas. Moreover, relapse compulsive buying episodes were identified as non-planned purchases associated with impulsive urges and negative urgency, followed by feelings of guilt and a loss of control independent of the amount of money spent once the treatment had begun. If a patient skipped three therapy sessions without informing the therapist, they were defined as dropout. In contrast, adherence to treatment was defined as patient compliance to therapeutic guidelines, including performing inter-sessions tasks such as recording and controlling their spends, and avoiding risky situations. Poor compliance was considered to be when a breach between these inter-sessions tasks occurred. 

### 2.3. Instruments

#### 2.3.1. Diagnostic Criteria for CBSD 

These criteria for CBSD according to McElroy et al. [24] have received wide acceptance in the research community, although their reliability and validity have not yet been determined [25]. It is worth noting that no formal diagnostic criteria for CBSD have been accepted for the Diagnostic and Statistical Manual of Mental Disorders, Fifth Edition (DSM-5) [26], nor for the International Classification of Diseases, Eleventh Revision (ICD-11) [27]. At present, it is recommended that CBSD diagnosis should be determined via detailed face-to-face interviews which explore “buying attitudes, associated feelings, underlying thoughts, and the extent of preoccupation with buying and shopping” [28].

#### 2.3.2. Symptom Checklist-Revised (SCL-90-R)

The SCL-90-R [29] is a 90-item questionnaire utilized to assess psychological distress and psychopathology in 9 symptom dimensions: somatization, obsessive–compulsive, interpersonal sensitivity, depression, anxiety, hostility, phobic anxiety, paranoid ideation, and psychoticism. The global score (Global Severity Index [GSI]) is a widely used index of psychopathological distress. In this study, the adapted version for the Spanish population was used [30]. Cronbach’s alpha (a) in the sample of this work was in the good to excellent range (The second table in this manuscript includes a-values for each scale).

#### 2.3.3. Other Sociodemographic and Clinical Variables

Specific CBSD behavior information was collected, such as the age of onset of CBSD, the mean and total maximum of monetary investment in a single episode of buying, and the total accumulated debt. Other variables associated with buying were assessed through a semi-structured face-to-face clinical interview, including demographic, clinical, and social/family variables [19]. The social position was calculated with the Hollingshead’s definition index [31], which provides a measure of the positions that individual or nuclear families occupy in the status structure of the society, based on different factors: education, occupation, marital status, and employment status. The family support for CBSD related problems was registered according to the patients’ perceptions: (a) lack of family support; (b) the family offers some form of support but patients perceive it as insufficient; and (c) family support is perceived by patients as full or sufficient. Substance use was directly informed by the patients in three items codified in a binary scale (no/yes) for tobacco use, alcohol use, and the use of other illegal drugs (this measurement did not allow differences between the non-problematic use and problematic abuse of substances). 

### 2.4. Procedure

The data analyzed in this work correspond to the assessment at baseline (before the CBT) and the outcomes reported by the patients during the treatment. With the arrival at the unit, all of the patients completed the diagnostic and screening tools, as well as the semi-structured clinical interview. Next, during the treatment, patients reported in each session the adherence to the therapy guidelines and the presence of relapses.

### 2.5. Statistical Analysis

Statistical analysis was carried out with Stata17 for Windows [32]. A comparison between the groups was performed with chi-square tests (|^2^) for categorical variables (the exact *p*-value was obtained for comparisons with expected counts less than 5) and with a *t*-test for independent samples for quantitative variables (the Kolmogorov–Smirnov test of normality required for these procedures reported non-significant results). 

Survival analyses were used to explore the rate of dropout, as well as relapses. The Kaplan–Meier (product–limit) estimator estimated the cumulative survival functions, and the Log Rank test compared the curves between the groups. The survival function is a method used to measure the probability of patients “living” (surviving without the presence of the outcome, for example, without dropouts) for a certain amount of time (in the study, during the treatment). One of the most relevant advantages of this procedure is allowing for the possibility of modeling censored data, which occurs if patients withdraw from the study (arrive alive at the end of the follow-up, or are lost to the follow-up without event occurrence at last measurement time) [33].

In this study, the effect size for the comparison between the groups was estimated with Cramer-V for the χ^2^, and with the Cohen-*d* coefficient for the difference between the means [34]. Effect size interpretation was based on Fergusson’s guidelines [35]: recommended minimum effect size representing a practically significant effect (RMPE) for |*d*| > 0.41 or *C-V* > 0.20, moderate effect for |*d*| > 1.15 or *C-V* > 0.50, and strong for |*d*| > 2.70 or *C-V* > 0.80.

The Finner method procedure was used to control the increase in Type-I errors due the use of multiple null-hypothesis tests. This is a stepwise multiple test procedure aimed to adjust *p*-values controlling the familywise error rate (FWER, defined as the probability that the statistical system makes at least *k* false rejections). When controlling the *k*-FWER, a fixed number of *k* − 1 of erroneous rejections is tolerated, and where all of the null hypotheses are equal, controlling the FWER at level α is equivalent to the problem of combining *p*-values to obtain a single testing for a null-hypothesis, which is at level α [36].

## 3. Results

### 3.1. Characteristics of the Participants

Table 1 displays the descriptive values for the sociodemographic variables, as well as the comparison between the groups. Within the total sample, most participants in the study reported primary or secondary education levels, married civil status, belonged to mean-low or low social position levels, were employed, and had a mean age of M = 43.0 years (SD = 10.6). Differences between the groups (significant results in the null-hypothesis tests or effect size higher than RMPE) were obtained for all of the sociodemographic, with the exception of employment status: this group of mothers reported a higher proportion of participants who were married, with lower social indexes, and with an older mean age.

### 3.2. Comparison between the Groups at Baseline

The psychological state, the CBSD related variables, and the prevalence of substance use and abuse registered at baseline are displayed in Table 2, as well as the comparison between the groups. No association between psychopathology and motherhood was observed (mean scores in the SCL-90-R were statistically equal between the groups, and effect sizes were lower than RMPE). The group of women with children reported an older age of onset of CBSD, a lower amount of Euros spent per compulsive buying episode (considering the maximum money spent reported), lower risk of alcohol use, and higher likelihood of family support for the CBSD (partial or complete).

### 3.3. Comparison between the Groups for the CBT Outcomes during the Treatment

The number of completed treatment sessions was in the range of 1 to 12 (mean = 8.8, SD = 4.1). The first grouped bar chart in Figure 1 represents the risk of dropout during the treatment (calculated for the total sample). The other grouped bar charts correspond to the risk of relapses, as well as the report of poor compliance of the treatment guidelines (estimated for the patients who completed the treatment). These results evidence that being a mother was associated with better CBT outcome, and concretely associated with lower risk of relapse (although null-hypothesis tests achieved non-significant results for the likelihood of relapses, effect sizes were higher than RMPE). 

Within the subsample of the patients who completed the treatment, the number of relapses during the treatment plan was also lower among women with children (M = 0.73 [SD = 1.68] versus M = 1.88 [SD = 2.73], t = 3.29, df = 44, *p* = *0*.002, *|d|* = 0.50). The total amount of money spent (Euros) reported in the relapses registered during the intervention was also lower (M = 68.5 [SD = 20.84] versus M = 152.8 [SD = 28.83], t = 7.73, df = 44, *p* < 0.001, *|d|* = 0.33).

### 3.4. Survival Analysis

Figure 2 includes the cumulative survival functions estimated for the groups of women with and without children, for the rate of dropout, and for relapses (the study of dropout was performed in the total sample, while relapses were analyzed in the subsample of the patients who completed the treatment). No significant results were obtained in the Log Rank tests for the rate of dropout, while mothers achieved a survival time without relapses which was longer than in women without children. 

## 4. Discussion

The aims of this study were twofold: (1) to compare the sociodemographic and clinical profile of women with CBSD who are mothers with those women who are childless; and (2) to determine whether motherhood has a positive or negative impact on the treatment response of these women by assessing dropout, relapse, and compliance rates. 

Regarding sociodemographic differences between groups, the mothers group reported an older mean age, lower social indexes, and a higher likelihood of being married. Marriage, as well as support and reciprocity with family members, along with other factors, has been considered a key developmental turning point that has enormous power as a mechanism of change for those seeking recovery [37]. Therefore, in addition to the specific role of motherhood, marriage could have played a highly relevant role in recovery. Sociodemographic differences between the two groups may have generated a certain bias in the results; as such, they should be interpreted with caution.

No association between psychopathology and motherhood was observed. The group of women with children reported an older age of onset of the buying-related problems, a lower amount of Euros spent per compulsive buying episode (considering the maximum money spent for episode reported), a lower risk of alcohol use, and a higher likelihood of family support for the CBSD. Previous studies have suggested that family support is associated with improved psychological and social health, along with a greater likelihood of treatment completion and sustained abstinence post-treatment [38,39,40]. Therefore, it is possible that, to some extent, these lower levels of clinical aspects associated with CBSD in the mothers’ group are due to the fact that they reported greater family support.

In accordance with our hypothesis, being a mother was also associated with better treatment outcomes, specifically with a lower risk of relapse and a lower proportion of poor treatment compliance. Moreover, within the subsample of the patients who completed the treatment, the number of relapses during the treatment plan was also lower among women with children, as well as the total number of Euros spent in the relapse episode. These results would support previous proposals, which consider motherhood as the most relevant internal facilitator in the field of addictions when it comes to seeking and engaging treatment and maintaining abstinence [41,42,43]. The desire to be a good mother and to break the generational cycle of addiction may, therefore, be more powerful than the challenges for recovery that mothers with addictions have to face, such as environmental cues (being with people or in places that normalize the addictive behavior), as well as difficulties in coping with stress, negative emotions, and interpersonal problems [44].

### 4.1. Research and Clinical Implications

As the results of the present study suggest, motherhood appears to be a possible internal facilitating state for recovery and treatment adherence for women with CBSD. Overall, motherhood appears to reduce the risk of relapse. Therefore, at the clinical level, motherhood and associated factors should be explored, with the aim of promoting greater adherence to treatment. On the other hand, for those women who do not have children, the presence of other factors that may promote a good response to treatment should be further examined. At research level, this study is an open door to explore the role of motherhood in other behavioral addictions in depth. 

### 4.2. Limitations and Future Research 

The present study has several limitations. First, some likely related variables, including the number and age of the children, and other possible treatment barriers were not taken into account. Future studies could explore whether these factors may play a determining role in the treatment response. Second, women who did not initiate treatment were not counted, and a waitlist control group was not included. Future studies could explore the percentage of pre-treatment dropout and its possible association with motherhood. Third, the reason for dropouts and relapses was not assessed; as such, it is not possible to determine whether they were related to maternal causes. Moreover, the accuracy measuring treatment response was limited, since there was no quantitative outcome variable for CBSD. Fourth, this study analyzed the data registered during the intervention; therefore, it is unknown how the compulsive buying behavior progressed during the follow-up after the treatment. Further research should assess the medium- and long-term effects after the intervention, along with the potential association with motherhood. Fifth, the fact that the same therapists performed the assessment and the treatment may represent a bias in the reported outcomes. For example, therapists may have expected mothers to perform better in treatment, which may have changed their treatment style, affecting the reporting outcomes. Finally, future studies could replicate the study with a larger sample, with women with other addictions (both behavioral and substance), as well as including fathers, to explore whether the weight of motherhood and fatherhood in recovery are similar.

## 5. Conclusions

The results of the present study suggest that motherhood is associated with a less severe clinical profile of women with CBSD, as well as with a better response to treatment (lower risk of relapse, as well as a lower risk of non-adherence to therapy guidelines). The results support the theoretical proposal that considers motherhood as an internal facilitating factor for recovery and treatment adherence of mothers with mental disorders. However, given the variability of psychological processes related to potential protective and risk factors in the context of motherhood and CBSD, more research on the potential mechanisms is needed to individualize and optimize treatment for women with CBSD.

## Figures and Tables

**Figure 1 ijerph-19-07075-f001:**
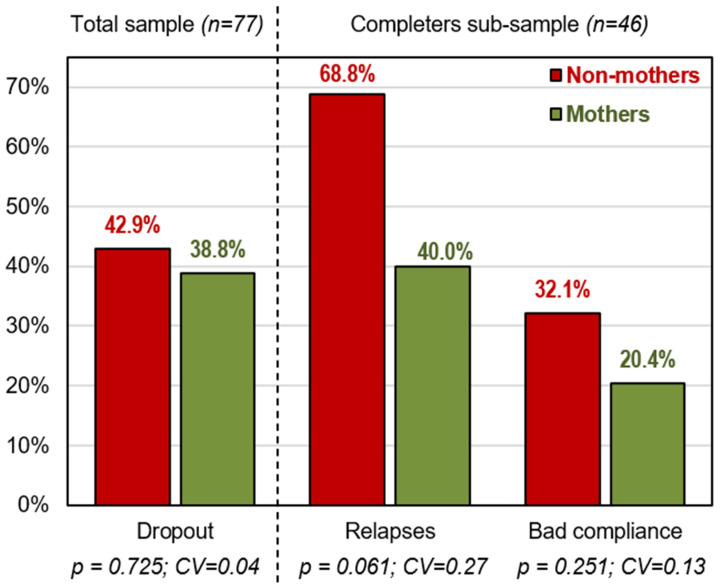
Presence of the therapy outcomes in the study.

**Figure 2 ijerph-19-07075-f002:**
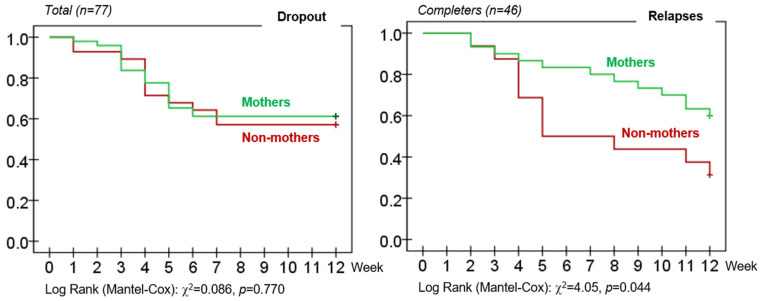
Cumulative survival function for the rate of dropout and relapses.

**Table 1 ijerph-19-07075-t001:** Descriptive of the sample.

		Total (*n* = 77)		Non-Mothers (*n* = 28)		Mothers (*n* = 49)			
		*n*	*%*	*n*	*%*	*n*	*%*	*p*	*C-V*
Education level	Primary	29	37.7%	6	21.4%	23	46.9%	0.077	**0.26 ^†^**
Secondary		29	37.7%	14	50.0%	15	30.6%		
University		19	24.7%	8	28.6%	11	22.4%		
Civil status	Single	27	35.1%	21	75.0%	6	12.2%	**<0.001 ***	**0.64 ^†^**
Married-in couple		40	51.9%	5	17.9%	35	71.4%		
Divorced-separated		10	13.0%	2	7.1%	8	16.3%		
Social index	Mean-high or high	17	22.1%	6	21.4%	11	22.4%	**<0.017**	**0.37 ^†^**
Mean		7	9.1%	5	17.9%	2	4.1%		
Mean-low		24	31.2%	12	42.9%	12	24.5%		
Low		29	37.7%	5	17.9%	24	49.0%		
Employment	Unemployed	35	45.5%	12	42.9%	23	46.9%	0.729	0.04
Employed		42	54.5%	16	57.1%	26	53.1%		
		*Mean*	*SD*	*Mean*	*SD*	*Mean*	*SD*	*p*	*|d|*
Age (years-old)		43.03	10.59	38.04	9.31	45.88	10.30	**0.001 ***	**0.80 ^†^**

Note. *SD*: standard deviation. * Bold: significant comparison (0.05 level). ^†^ Bold: effect higher than the recommended minimum effect size representing a practically significant effect.

**Table 2 ijerph-19-07075-t002:** Comparison of the clinical profile.

		Non-Mothers (n = 28)		Mothers (n = 49)			
**Psychopathology (SCL-90-R)**	*α*	*Mean*	*SD*	*Mean*	*SD*	*p*	|d|
Somatization	0.907	1.64	1.17	1.75	1.09	0.685	0.10
Obsessive/compulsive	0.898	2.12	1.09	1.83	1.04	0.267	0.28
Interpersonal sensitive	0.871	1.77	1.05	1.53	1.00	0.356	0.23
Depressive	0.914	2.38	1.12	2.28	1.07	0.711	0.09
Anxiety	0.908	1.76	1.06	1.70	1.20	0.841	0.05
Hostility	0.860	1.56	1.25	1.23	0.92	0.212	0.30
Phobic anxiety	0.844	1.09	1.15	0.97	1.03	0.638	0.12
Paranoid Ideation	0.808	1.60	1.04	1.33	0.94	0.275	0.27
Psychotic	0.831	1.29	0.95	1.24	0.97	0.828	0.05
GSI score	0.981	1.75	0.92	1.65	0.91	0.655	0.11
PST score	0.981	60.32	21.19	57.04	21.47	0.541	0.15
PSDI score	0.981	2.41	0.76	2.43	0.66	0.923	0.02
**Onset-duration buying problems**		*Mean*	*SD*	*Mean*	*SD*	*p*	|d|
Age of onset of buying problems		28.53	10.74	37.94	12.22	**0.001 ***	**0.82 ^†^**
Duration of buying problems		7.18	6.90	8.38	8.19	0.517	0.16
**Buying related variables**		*n*	*%*	*n*	*%*	*p*	C-V
Max. money spent/episode	Less 50 €	10	35.7%	27	55.1%	0.235	**0.23 ^†^**
50 to 100 €		1	3.6%	0	0.0%		
100 to 300 €		6	21.4%	7	14.3%		
More than 300 €		11	39.3%	15	30.6%		
Mean money spent/episode	Less 50 €	21	75.0%	39	79.6%	0.580	0.16
50 to 100 €		4	14.3%	3	6.1%		
100 to 300 €		2	7.1%	6	12.2%		
More 300 €		1	3.6%	1	2.0%		
Debts due to buying behaviors	No	18	64.3%	22	44.9%	0.101	0.19
Yes		10	35.7%	27	55.1%		
Family support	No	3	10.7%	3	6.1%	0.082	**0.24 ^†^**
Partial		2	7.1%	13	26.5%		
Complete		23	82.1%	33	67.3%		
**Prevalence substances use/abuse**		*n*	*%*	*n*	*%*	*p*	C-V
Tobacco		12	42.9%	14	28.6%	0.202	0.15
Alcohol		2	7.1%	0	0.0%	**0** **.042 ***	**0.22 ^†^**
Other illegal drugs		1	3.6%	1	2.0%	0.690	0.05

Note. SCL-90-R: Symptom Checklist, Revised; GSI: Global Severity Index; PST: Positive Symptom Total; PSDI: Positive Symptom Distress Index; α: Cronbach’s alpha in the study. *SD*: standard deviation. * Bold: significant comparison (0.05 level). ^†^ Bold: higher effect than the recommended minimum effect size representing a practically significant effect.

## Data Availability

The data will be available through a direct request to the authors, who will evaluate the type of information requested with the Clinical Research Ethics Committee at the University Hospital of Bellvitge. Due to the clinical nature of the sample analyzed and the fact that the patients signed a consent form for their data to be stored and kept by the hospital center, it is not possible to transfer them to an open data repository.

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
