# Peer review of "Motherhood and Treatment Outcome in Female Patients with Compulsive Buying–Shopping Disorder"

_ijerph, 2022, doi:10.3390/ijerph19127075_

Round 1
Reviewer 1 Report
The aim of this study was to compare CBT for compulsive buying in mothers versus non-mothers. I am not aware of another study like this, and think it is a valuable and novel addition to the literature on compulsive buying. Although this is a pilot study, there are some interesting findings that will hopefully lead to more research in this area.
Introduction
- I thought the introduction was concise and very clear. But it would be good to review previous literature on the efficacy on compulsive buying treatment more thoroughly (rather than just giving a couple citations in the methods).
Methods
- Please specify which structured interview was used to diagnose compulsive buying
- Please specify comorbidities in participants and whether there were any significant differences in comorbid mental health disorders between mothers and non-mothers.
- Please specify average number (and min and max) of sessions completed
- Please interpret effect sizes with Ferguson’s (2009) guidelines.
- Were all assumptions met for all statistical tests run in this study? If so, please note this in the data analysis section.
Results
- In table 2, I do not understand what social index indicates. Please explain this further.
- In table 2, please explain what is meant by partial and complete family support. Were there criteria used to categorise participants into each label?
- Please separate substance use and substance abuse statistics in Table 2 – compiling them does not seem appropriate. If possible, it would also be good to explain what criteria were used to diagnose substance abuse, and what defines substance use in your sample – ie 1 drink per week, or 5 drinks per week?
- In section 3.3, please clarify whether you mean relapses after treatment, or relapses during
- In section 3.3, please report df and Cohens d for the t-tests, as well as SD and n for the groups compared, to give a better idea of whether the tests were adequately powered.
- The survival analyses are very interesting, and the figures look great. However, they would be more informative if the methods section specified what was done during each week of the treatment program. It would be informative to see if relapse or dropout were associated with something in the treatment protocol (i.e. the start of ERP).
- Was there a difference between number of participants in each group who were fully recovered posttreatment?
Discussion
- The outcomes of the study don’t seem to be accurately reported. In the 4th paragraph of the discussion, you write that being a mother was also associated with better treatment outcomes, specifically with lower risk of relapse and lower proportion of bad treatment compliance. But in Figure 1, it says that there is no significant differences between risk of relapse and risk of bad compliance (p = .061 and p=.251, respectively). Please note that these differences were not significant.
- Were there any barriers for treatment reported by mothers in this study? Perhaps this is something that can also be assessed in future studies.
- Please specify that a limitation of this study is absence of control group (ie waitlist control).
- Another limitation is that the same therapists did assessment and treatment, which may bias their treatment and reporting of outcomes. Ie Therapists may have expected mothers to do better in treatment, which may have changed their treatment style, and reporting of outcomes.
- Please note in the limitations that there was no quantitative outcome variable for compulsive buying which limits the accuracy in measurement of treatment response
Minor comments
- Introduction – paragraph 2. Full stop needed before “Finally, it is necessary to consider various…”
- You should refer to this disorder as “Compulsive Buying-Shopping Disorder” to be consistent with the ICD-11 and other recent literature.
- I assume “bets/episode” actually means “money spent/episode” in section 3.2 and Table 2? Please fix this as bets/episode does not make sense for compulsive buying.
- Explain what GSI, PST and PSDI stands for in the table 2 caption.
- I think it would be better to label the groups as mothers and non-mothers rather than children- and children+.
Author Response
REVIEWER 1
The aim of this study was to compare CBT for compulsive buying in mothers versus non-mothers. I am not aware of another study like this, and think it is a valuable and novel addition to the literature on compulsive buying. Although this is a pilot study, there are some interesting findings that will hopefully lead to more research in this area.
Introduction
- I thought the introduction was concise and very clear. But it would be good to review previous literature on the efficacy on compulsive buying treatment more thoroughly (rather than just giving a couple citations in the methods).
Response: We thank the reviewer for their positive feedback. We consider the information on CBSD treatment suggested by the reviewer to be very appropriate and have added the following paragraph to the introduction section:
“Multiple approaches have been proposed for CBSD both pharmacological (i.e. citalopram, escitalopram, and fluvoxamine) and psychological (i.e. psychodynamic psychotherapy, behavioral approaches, and cognitive-behavioral interventions). Of all these interventions, cognitive-behavioral therapy (CBT) appears to be particularly effective reducing the associated distress of CBSD, with long-lasting effects [16]”.
Methods
- Please specify which structured interview was used to diagnose compulsive buying
Response: We find the reviewer's question very interesting. The structured clinical interview that was used in the present study includes the following sections to comprehensively diagnose CBSD.
Topographical analysis:
- Motor, cognitive, and physiological responses associated with shopping episodes. Frequency, intensity, and duration of shopping episodes. Maximum expenditure per episode, and average expenditure.
- To evaluate the occurrence of shopping episodes, in physical store and online (determining the usual shopping platform: PC, smartphone and/or tablet), and to determine the specific websites where the patient shops, forms of payment, and most frequently purchased items.
Functional analysis:
- Antecedents: Triggers (external, such as having money, window displays, sales, offers, advertisements for new items, etc. and internal, such as negative emotional states).
- Motor, physiological, cognitive, and emotional responses.
- Short-term and long-term consequences.
- To analyze what the patient does with the purchased objects: accumulate, return them, and/or give them away.
In the manuscript we included the following information:
“Specific CBSD behavior information was collected, such as the age of onset of CBSD, the mean and total maximum of monetary investment in a single episode of buying and the total accumulated debt. Other variables associated with buying were assessed through a semi structured face-to-face clinical interview, including demographic, clinical, and social/family variables [17].”
- Please specify comorbidities in participants and whether there were any significant differences in comorbid mental health disorders between mothers and non-mothers.
Response: Thank you for pointing this out. This study used the SCL-90-R scales to assess the presence of 90 potential comorbid psychopathological symptoms, structured in nine first-order dimensions (somatization, obsessive-compulsive, interpersonal sensitivity, depression, anxiety, hostility, phobic anxiety, paranoid ideation and psychoticism). This tool also provides 3 global indexes, which provide the total symptoms identified and the global psychopathology distress. The comparison between the SCL-90-R scales is displayed in Table 2, as well as the comparison for the substances use-abuse. According to the results of this Table, the results section indicates that:
“No association between psychopathology and motherhood was observed (mean scores in the SCL-90-R were statistically equal between the groups, and effect sizes were lower than RMPE). The group of women with children reported older age of onset of CBSD, lower amount of euros spent per compulsive-buying episode (considering the maximum money spent reported), lower risk of alcohol use and higher likelihood of family support for the CBSD (partial or complete).”
- Please specify average number (and min and max) of sessions completed
Response: Thank you for this recommendation. We have reported this new information in the results section: “The number of completed treatment sessions was into the range of 1 to 12 (mean=8.8, SD=4.1).”
- Please interpret effect sizes with Ferguson’s (2009) guidelines.
Response: We thank for this sound comment. We have now used the thresholds provided in the study published by Christopher J. Ferguson (2009, DOI: 10.1037/a0015808) for defining the recommended minimum effect size representing a “practically” significant effect (see statistical analysis section):
“Effect size interpretation was based on Fergusson’s guidelines [33]: recommended minimum effect size representing a practically significant effect (RMPE) for |d|>0.41 or C-V>0.20, moderate effect for |d|>1.15 or C-V>0.50, and strong for |d|>2.70 or C-V>0.80.”
- Were all assumptions met for all statistical tests run in this study? If so, please note this in the data analysis section.
Response: Thank you for this interesting report. We have now clarified that the exact p-value was obtained for the chi-square procedures with expected counts less than 5, and that the Kolmogorov-Smirnov tests for normality required for the T-TEST procedures reported non-significant results:
“Comparison between the groups was done with chi-square tests (c2) for categorical variables (exact p-value was obtained for comparisons with expected counts less than 5) and with T-Test for independent samples for quantitative variables (the Kolmogorov-Smirnov tests of normality required for these procedures reported non-significant results).”
Results
- In table 2, I do not understand what social index indicates. Please explain this further.
Response: The social position index used in the study was calculated with the Hollingshead definition (2011). We have now clarified in the measures section that this is a measure of the positions individual or nuclear families occupy in the status structure of the society, based on the different factors: education, occupation, marital status and employment status.
- In table 2, please explain what is meant by partial and complete family support. Were there criteria used to categorize participants into each label?
Response: We appreciate the Reviewer’s constructive comment. During the clinical interview, the patients are asked if the family members offers support for the problems caused by compulsive buying related problems. Their response is then codified in 3 categories, according to the evaluation made by the own patients: 1) the family does not offer any support; (b) the family offers some form of support, but patients perceive it as insufficient; and (c) family support is perceived by patients as full or sufficient. We have clarified this issue in the results section.
“The family support for the CBSD related problems was registered, according to the patients’ perceptions: 1) lack of family support; (b) the family offers some form of support, but patients perceive it as insufficient; and (c) family support is perceived by patients as full or sufficient.”
- Please separate substance use and substance abuse statistics in Table 2 – compiling them does not seem appropriate. If possible, it would also be good to explain what criteria were used to diagnose substance abuse, and what defines substance use in your sample – ie 1 drink per week, or 5 drinks per week?
Response: Thank you for this constructive remark. In this study, we have not used standardized measurement tools or definitions to identify the presence/absence of substance related disorders. During the clinical interview, the patients only reported the use of tobacco, alcohol, or other illegal drugs (the frequency and potential harmful was not measured). This is the reason for using the label “substances use-abuse”, since it was not possible to differentiate the non-problematic versus problematic consumption. We have clarified this in the measures section:
“Substance use was directly informed by the patients, in three items codified in a binary scale (no/yes) for the tobacco use, the alcohol use and the other illegal drugs (this measurement did not allow differentiate between the non-problematic use and problematic abuse of substances).”
- In section 3.3, please clarify whether you mean relapses after treatment, or relapses during
Response: We appreciate this thoughtful feedback provided from Reviewer. We have clarified that the therapy outcomes reported in the study were measured during the treatment (in the title of the section 3.3 and also in the description of the results referred to the relapses).
- In section 3.3, please report df and Cohens d for the t-tests, as well as SD and n for the groups compared, to give a better idea of whether the tests were adequately powered.
Response: Thank you for highlighting this. We have now included these numerical results in the manuscript.
“Within the subsample of the patients who completed the treatment, the number of relapses during the treatment plan was also lower among women with children (M=0.73 [SD=1.68] versus M=1.88 [SD=2.73], df=44, p=.002, |d|=0.50) and the total amount of money spent (euros) reported in the relapses registered during the intervention was also lower (M=68.5 [SD=20.84] versus M=152.8 [SD=28.83], df=44, p<.001, |d|=0.33).”
- The survival analyses are very interesting, and the figures look great. However, they would be more informative if the methods section specified what was done during each week of the treatment program. It would be informative to see if relapse or dropout were associated with something in the treatment protocol (i.e. the start of ERP).
Response: We thank for this sound comment. In the limitations section we had clarified that the reason for dropouts and relapses was not assessed, so it was not possible to determine whether they were related to maternal causes.
- Was there a difference between number of participants in each group who were fully recovered posttreatment?
Response: Thank you for this constructive remark. We totally agree with the reviewer
We fully agree with the reviewer on the relevance of knowing to what extent the results obtained after the intervention were maintained during the follow-up. Unfortunately, at the time of this research no information was available regarding the post-treatment, and for this reason it was not possible to provide more information about the medium-long term effects of the CBT, nor its potential differences based on the independent variable in this research (motherhood). We have now included this new content in the limitations section:
“This study analyzed the data registered during the intervention, therefore it is unknown how the compulsive buying behavior progressed during the follow-up after the treatment. Further research should assess the medium- and long-term effects after the intervention and the potential association with motherhood.”
Discussion
- The outcomes of the study don’t seem to be accurately reported. In the 4th paragraph of the discussion, you write that being a mother was also associated with better treatment outcomes, specifically with lower risk of relapse and lower proportion of bad treatment compliance. But in Figure 1, it says that there is no significant differences between risk of relapse and risk of bad compliance (p = .061 and p=.251, respectively). Please note that these differences were not significant.
Response: The reviewer is correct in pointing out that in Figure 1 there are no statistically significant differences, but when analyzing the group of patients who do complete treatment, it can be observed that mothers achieved a survival time without relapses higher than women without children.
- Were there any barriers for treatment reported by mothers in this study? Perhaps this is something that can also be assessed in future studies.
Response: We consider the information suggested by the reviewer to be very relevant, but unfortunately this data was not recorded in our study. We have included the following limitation:
“First, some likely related variables including the number and age of the children, and other possible treatment barriers were not taken into account. Future studies could explore whether these factors may play a determining role in the treatment response.”
- Please specify that a limitation of this study is absence of control group (ie waitlist control).
Response: We agree with this comment, and we have added the following information:
“Second, women who did not initiate treatment were not counted, and a waitlist control group was not included”.
- Another limitation is that the same therapists did assessment and treatment, which may bias their treatment and reporting of outcomes. Ie Therapists may have expected mothers to do better in treatment, which may have changed their treatment style, and reporting of outcomes.
Response: We have added the following limitation:
“Fifth, the fact that the same therapists performed the assessment and the treatment, may represent a bias in the reported outcomes. For example, therapists may have expected mothers to do better in treatment, which may have changed their treatment style, affecting the reporting outcomes.”
- Please note in the limitations that there was no quantitative outcome variable for compulsive buying which limits the accuracy in measurement of treatment response
Response: We have added the following limitation:
“Moreover, the accuracy measuring treatment response was limited, since there was no quantitative outcome variable for CBSD.”
Minor comments
- Introduction – paragraph 2. Full stop needed before “Finally, it is necessary to consider various…”
Response: We thank the reviewer for their eye for detail. We have added the missing full stop.
- You should refer to this disorder as “Compulsive Buying-Shopping Disorder” to be consistent with the ICD-11 and other recent literature.
Response: In agreement with reviewer’s comment, we have modified “compulsive buying disorder” by “compulsive buying-shopping disorder”.
- I assume “bets/episode” actually means “money spent/episode” in section 3.2 and Table 2? Please fix this as bets/episode does not make sense for compulsive buying.
Response: In agreement with reviewer’s suggestion, this mistake has been rectified. We have modified “bets/episode” by “money spent/episode”.
- Explain what GSI, PST and PSDI stands for in the table 2 caption.
Response: In agreement with reviewer’s suggestion, we have added the following information in table 2 captation:
“SCL-90-R: Symptom Checklist-Revised; GSI: Global Severity Index; PST: Positive Symptom Total; PSDI: Positive Symptom Distress Index”.
- I think it would be better to label the groups as mothers and non-mothers rather than children- and children+.
Response: We have modified “children-“ by “non-mothers” and “children+” by “mothers”.
Reviewer 2 Report
This article makes a good contribution to the literature. Overall, I found the topic very interesting and the writing quite strong. With that said, it is in the spirit of strengthening the manuscript that I offer the following questions/ comments/ recommendations:
The introduction is poor and should be more developed. The authors pay very little attention to some aspects in introduction that need to be improved. It should be enriched with the description of some theoretical model on the determinants and benefits of parenting (or motherhood). Also, while being a compulsive buyer is a mental health issue it certainly does not have the same implications for family life and childcare as other mental illnesses or addictions (being an alcoholic, heroin user, major depression, etc...).
Please correct the 1st sentence of the introduction, motherhood is not a factor. It’s a experience of being a mother, a state, a identity and so on….
2nd paragraph “On the other hand, successful recovery usually has multiple aspects, such as im-proved self-care”. Please specify what recovery you are talking about.
Methods and results are well presented. However, in the Procedure, please specify more detail about the data collection process. Also Tables 1 and 2 need to be edited.
Discussion: You should discuss further the possible bias caused by sociodemographic differences between the two groups.
In what ways can future scholars build on your work?
How might the findings of your study inform the work of clinicians/practitioners? How can clinicians use the findings of your study?
Author Response
REVIEWER 2
This article makes a good contribution to the literature. Overall, I found the topic very interesting and the writing quite strong. With that said, it is in the spirit of strengthening the manuscript that I offer the following questions/ comments/ recommendations:
The introduction is poor and should be more developed. The authors pay very little attention to some aspects in introduction that need to be improved. It should be enriched with the description of some theoretical model on the determinants and benefits of parenting (or motherhood). Also, while being a compulsive buyer is a mental health issue it certainly does not have the same implications for family life and childcare as other mental illnesses or addictions (being an alcoholic, heroin user, major depression, etc...).
Response: It may seem that compulsive shopping is a problem with less impact and negative consequences in the life of the individual who suffers from it, if we compare it with other mental disorders. However, the clinical reality is quite different. Usually, patients seeking treatment present severe clinical pictures, with significant consequences on their lives. All of them have accumulated an important number of debts that they cannot face, they have real problems at family and couple level, so many that it is frequent to see that they are in the process of divorce. In addition, it is a disorder with high comorbidity with anxious-depressive disorders, eating disorders, obsessive-compulsive disorder, hoarding, etc. For all these reasons, the impact on the well-being of patients and their families is significant.
In order to enrich the introduction, as suggested by the reviewer, we have added the following paragraphs:
“Previous studies have suggested that motherhood can be a motivating factor in recovering from substance use disorder, although it can also be a barrier due to fear of possible negative consequences, such as loss of child custody [11,12]. However, this aspect has not been extensively studied in behavioral addictions.”
“Multiple approaches have been proposed for CBSD both pharmacological (i.e. citalopram, escitalopram, and fluvoxamine) and psychological (i.e. psychodynamic psychotherapy, behavioral approaches, and cognitive-behavioral interventions). Of all these interventions, cognitive-behavioral therapy (CBT) appears to be particularly effective reducing the associated distress of CBSD, with long-lasting effects [18].”
Please correct the 1st sentence of the introduction, motherhood is not a factor. It’s a experience of being a mother, a state, a identity and so on….
Response: We completely agree with this comment and we have modified “factor” to “state”.
2nd paragraph “On the other hand, successful recovery usually has multiple aspects, such as im-proved self-care”. Please specify what recovery you are talking about.
Response: In accordance with the reviewer's suggestion, we have clarified the term "recovery". It now reads:
“On the other hand, mothers with mental illness’ successful recovery usually includes multiple aspects, such as improved self-care, involvement in treatment programs, and reconstruction of one's identity [4–6].”
Methods and results are well presented. However, in the Procedure, please specify more detail about the data collection process. Also Tables 1 and 2 need to be edited.
Response: We appreciate this helpful suggestion. We have now reviewed Tables 1 and 2, and we have also included the next content in the new 2.4 section (labeled “procedure”):
“The data analyzed in this work correspond to the assessment at baseline (before the CBT) and the outcomes reported by the patients during the treatment. At the arrival to the unit, all the patients completed the diagnostic and screening tools, as well as the semi-structured clinical interview. Next, during the treatment, patients reported in each session the accomplishment with the therapy guidelines and the presence of relapses.”
Discussion: You should discuss further the possible bias caused by sociodemographic differences between the two groups.
Response: In accordance with the reviewer's suggestion, we have incorporated the following paragraph in the discussion section:
“Marriage and support and reciprocity with family members, along with other factors, has been considered a key developmental turning point that has strong power as a mechanism of change for those seeking recovery [35]. Therefore, in addition to the specific role of motherhood, marriage could have played a highly relevant role in recovery. Sociodemographic differences between the two groups may have generated a certain bias in the results, so they should be interpreted with caution.”
In what ways can future scholars build on your work? How might the findings of your study inform the work of clinicians/practitioners? How can clinicians use the findings of your study?
Response: We consider the information suggested by the reviewer to be very appropriate. We have added the section “research and clinical implications”:
“4.1. Research and clinical implications
As the results of the present study suggest, motherhood appears to be a possible internal facilitating state for recovery and treatment adherence for women with CBSD. Overall, motherhood seems to reduce the risk of relapse. Therefore, at the clinical level, motherhood and associated factors should be explored with the aim of promoting greater adherence to treatment. On the other hand, for those women who do not have children, the presence of other factors that may promote a good response to treatment should be further examined. At research level, this study is an open door to explore the role of motherhood in depth in other behavioral addictions.”
Round 2
Reviewer 1 Report
Thank you for incorporating my feedback. The paper looks a lot better. Here are two small things I noticed that need to be fixed:
1. Please provide a t-statistic for the comparison of number of relapses between mothers and non-mothers during treatment.
2. Figure 1 and 2 still mention children- and children+ group labels. Please make the labels consistent with the rest of the paper.
Author Response
REVIEWER 1
Thank you for incorporating my feedback. The paper looks a lot better. Here are two small things I noticed that need to be fixed:
- Please provide a t-statistic for the comparison of number of relapses between mothers and non-mothers during treatment.
- Figure 1 and 2 still mention children- and children+ group labels. Please make the labels consistent with the rest of the paper.
Reply: We thank the reviewer for their help in improving the article and for the positive feedback. We have provided the t-statistic and we have modified Figure 1 and 2.
Reviewer 2 Report
First, I would like to thank the authors for addressing my previous concerns/comments/questions. Second, I am generally pleased with the changes that have been made. So I recommend the publication of the manuscript.
Author Response
REVIEWER 2
First, I would like to thank the authors for addressing my previous concerns/comments/questions. Second, I am generally pleased with the changes that have been made. So I recommend the publication of the manuscript.
Reply: We thank the reviewer for their help in improving the article and for the positive feedback.